# Role of Pyroptosis in Intervertebral Disc Degeneration and Its Therapeutic Implications

**DOI:** 10.3390/biom12121804

**Published:** 2022-12-02

**Authors:** Jieya Luo, Yuxuan Yang, Xuan Wang, Xingyu Chang, Songbo Fu

**Affiliations:** 1The Clinical Medical College, Guizhou Medical University, Guiyang 550004, China; 2The First Clinical Medical College, Lanzhou University, Lanzhou 730000, China; 3Department of Endocrinology, First Hospital of Lanzhou University, Lanzhou 730000, China; 4Gansu Province Clinical Research Center for Endocrine Disease, Lanzhou 730000, China

**Keywords:** pyroptosis, intervertebral disc degeneration, NLRP3 inflammasome, nucleus pulposus

## Abstract

Intervertebral disc degeneration (IDD), a progressive and multifactorial pathological process, is predominantly associated with low back pain and permanent disability. Pyroptosis is a type of lytic programmed cell death triggered by the activation of inflammasomes and caspases. Unlike apoptosis, pyroptosis is characterized by the rupture of the plasma membrane and the release of inflammatory mediators, accelerating the destruction of the extracellular matrix (ECM). Recent studies have shown that pyrin domain-containing 3 (NLRP3) inflammasome-mediated pyroptosis in nucleus pulposus (NP) cells is activated in the progression of IDD. Furthermore, targeting pyroptosis in IDD demonstrates the excellent capacity of ECM remodeling and its anti-inflammatory properties, suggesting that pyroptosis is involved in the IDD process. In this review, we briefly summarize the molecular mechanism of pyroptosis and the pathogenesis of IDD. We also focus on the role of pyroptosis in the pathological progress of IDD and its targeted therapeutic application.

## 1. Introduction

Intervertebral disc (IVD) degeneration (IDD) is a leading cause of low back pain (LBP), affecting more than 80% of the population worldwide [1]. The global burden of disease study 2016 [2] ranked LBP as the leading cause of increased disability, with a lifetime prevalence of approximately 84%. In addition to lowering quality of life, IDD is also associated with a heavy socioeconomic burden. In the United States alone, healthcare expenditures for diseases related to IDD have reached USD 134.5 billion [3]. Notably, epidemiological surveys have reported a higher incidence of IDD in the elderly [4], postmenopausal women [5], and manual workers [6]. The pathogenesis of IDD is complex, involving an interplay of multiple factors which remain incompletely understood. It is generally believed that nucleus pulposus (NP) cell death [7], extracellular matrix (ECM) metabolic disorder [8], and secondary inflammation [9] are the three main causes of IDD. Macroscopically, IDD manifests as structural destruction and progressive dysfunction, including endplate (EP) sclerosis, osteophyte formation, and limited movement [10]. IDD can be improved with medical treatment in its early stages [11]. While the symptoms cannot be entirely relieved by conservative treatment, surgery is usually necessary to relieve the pain caused by the rupture or herniation of the IVD and to restore partial motor function. However, in essence, surgical treatment is still not an etiological treatment and fails to prevent or delay disease progression [12]. Additionally, its unclear curative effect, frequent complications, and high risk of recurrence cannot be ignored [13].

Initially observed in Shigella flexneri-infected macrophages in 1992 [14], pyroptosis was determined to be a form of proinflammatory cell death activated by caspase-1 in 2001 [15]. The term was coined from the Greek word pyro (fire or fever), reflecting its inflammatory nature, and ptosis (falling), meaning cell death. As it disrupts cell integrity features and belongs to a kind of programmed death driven by caspases, pyroptosis morphologically appears to be a combination of necrosis and apoptosis [16]. However, as a new type of cell death, pyroptosis has its specificity. In 2018, pyroptosis was defined as a form of cell death that was pore-formed by the gasdermin protein family and is often—but not always—activated by inflammatory caspase [17]. In recent years, pyroptosis has participated in many disease processes, including viral infection [18], tumors [19], and neurodegenerative diseases [20], etc. A broad set of studies have shown that NLRP3 inflammasome-mediated pyroptosis plays a significant role in IDD, and the inhibition of pyroptosis can alleviate IDD progression [21,22,23].

In this review, we discuss the role of pyroptosis in IDD progression based on the molecular mechanism of pyroptosis and the progression of IDD. Additionally, we summarize IDD therapeutic strategies targeting pyroptosis, providing a theoretical basis for further research.

## 2. Pyroptosis

### 2.1. Canonical Inflammasome-Induced Pyroptosis

The canonical inflammasome-induced pyroptosis pathway begins with inflammasome assembly [24] (Figure 1). As a multiprotein complex in the cytoplasm, the inflammasome can be divided into three parts according to its domains: pattern recognition receptors (PRRs), apoptosis-associated speck-like protein containing caspase recruitment domain (ASC), and pro-caspase-1. Among them, PRRs comprise AIM2-like receptors, nucleotide-binding oligomerization domain-like receptors (NLRs), and pyrin [25]. PRRs, so-called cytoplasmic sensors, can recognize some molecules and then deliver them inside the host. These molecules contain “pathogen-associated molecular patterns (PAMPs)”, including bacteria, fungi, viruses, etc., as “non-self” signatures, and “damage-associated molecular patterns (DAMPs)”, such as ATP, cholesterol, interleukin (IL)-1β, and IL-18 [26]. Acting as the “adhesive” of the inflammasome, ASC contains a pyrin domain (PYD) and a caspase activation and recruitment domain (CARD). Stimulated by PRRs, pro-caspase-1 is recruited through typical protein–protein interactions, including homotypic ASC-PYD and PRRs-PYD, as well as ASC-CARD and pro-caspase-1-CARD interactions [27]. Finally, performing as an “effector”, pro-caspase-1 is an inactive precursor of active caspase-1. After the inflammasome complex is assembled, it obtains proteolytic activity by self-activating pro-caspase-1 to caspase-1, which in turn cleaves pro-IL-1β and pro-IL-18 into their mature secreted forms [25]. On the other hand, caspase-1 cleaves gasdermin D (GSDMD) by aggregating and self-processing it into caspase-1 p33/p10 [28]. GSDMD is a member of the gasdermin protein family, which consists of two conserved domains: the N-terminal pore-forming domain (PFD) and the C-terminal repressor domain (RD). Under normal circumstances, GSDMD is found in a quiescent state as a result of the autoinhibition produced by the interaction between PFD and RD [29]. After being cleaved by caspase-1, PFD binds to phosphatidylinositol of the eukaryotic cell membrane and cardiolipin of the prokaryotic cell membrane, and then assembles into a hollow oligomer with a diameter of 10–14 nm [30]. Exceeding the limitation of the endosomal sorting complex required for transport (ESCRT)-dependent membrane repair will lead to osmotic cell swelling and membrane perforation, releasing IL-1β and IL-18 [31].

Of note, it is usually postulated that the NLRP3 inflammasome is a key effector molecule of pyroptosis due to its multiple layers of transcriptional and post-translational modifications [32]. The NLRP3 inflammasome is generally expressed at low levels and remains ubiquitinated. Conventionally, the activation of the NLRP3 inflammasome can be divided into two phases: the “priming phase” and the “activating phase” [33]. The essential task of the priming signal is to deal with the increased expression and the post-translational modification of the NLRP3 inflammasome. To be more specific, toll-like receptors (TLRs) or cytokine receptors, including IL-1 receptors and tumor necrosis factor (TNF)-α receptors, can activate the transcription factor NF-κB and upregulate the expression of NLRP3 and pro-IL-1β by identifying ligands (lipopolysaccharide (LPS), TNF-α, IL-1, etc.) individually [34]. Additionally, deubiquitination [35] and phosphorylation [36] are required in the priming step. The activating phase of the NLRP3 inflammasome is triggered by the intracellular disturbance caused by multiple agonists. Among them, ionic flux, reactive oxygen species (ROS), and mitochondrial dysfunction are considered to be central factors [33].

### 2.2. Noncanonical Inflammasome-Induced Pyroptosis

The noncanonical inflammasome-induced pyroptosis pathway is mediated by human caspase-4/5 and mouse homologous caspase-11. In this way, LPS, a component of Gram-negative bacteria, directly activates caspase-11/4/5, playing a similar function to caspase-1 [37]. These inflammatory caspases cleave GSDMD to incur pyroptosis [38]. However, instead of converting pro-IL-1β and pro-IL-18 directly, caspase-4/5/11 improves the maturation of IL-1β and IL-18 through the NLRP3-caspase-1 pathway [39]. K^+^ behaves as a bridge, connecting the noncanonical pathway with the NLRP3 inflammasome. While GSDMD activation triggers membrane perforation, the efflux of K^+^ engages in the NLRP3 inflammasome activation [40]. Another study found that, upon intracellular LPS stimulation, caspase-11 specifically modifies the pannexin-1 channel, resulting in ATP release. ATP, the NLRP3 agonist, acting through P2X7 receptors on the cell membrane, triggers K^+^ efflux [41].

### 2.3. Other Relevant Approaches

The discovery that apoptosis-associated caspases-3/8 induce pyroptosis by cleaving GSDME contributes to a deeper understanding of pyroptosis. It has previously been demonstrated that chemotherapy drugs such as mitoxantrone, actinomycin-D, and cisplatin can convert the apoptosis phenotype to caspase-3/GSDME-dependent pyroptosis. Interestingly, the type of cell death is relevant to the expression level of GSDME [42]. New insights into the crosstalk between GSDMD inhibition in caspase-3 and activation in caspase-8 may elucidate interesting mechanisms concerning pyroptosis [43]. Furthermore, caspase-independent pyroptosis has been found in some cells. Chimeric antigen receptor T (CAR-T) cells can release granzyme B to activate caspase-3 [44] and lymphocytes produce granzyme A, acting on GSDMB directly [45]. Elastase, the serine protease in neutrophils, can also cleave GSDMD [46]. Further efforts are required to shed more light on this underlying mechanism.

## 3. The Pathogenesis of IDD

IDD is considered an age-related disease. After going through a degenerative cascade, it ultimately results in the loss of structural integrity, limited movement, pain, and decreased quality of life [10]. Causative factors of IDD can be categorized into passive destruction, including a violent blow or postural abrasion, and progressive changes. The latter is associated with aging and external environmental factors such as smoking, obesity, inflammation, and oxidation, etc. [47]. All of the above play a pathogenic role by changing the structure and metabolism of the IVD. A normal IVD is an avascular and innervated airtight fibrocartilaginous tissue located between the vertebrae. It consists of three parts: the nucleus pulposus (NP), the annulus fibrosus (AF), and the endplate (EP) [48]. Centrally located, the NP is surrounded by the AF, with the outermost EP wrapping the two structures together. Together, they constitute a closed cushioning system against stress, which enables the IVD to absorb and disperse loads from the vertebrae [10].

When IDD occurs, one of the most striking changes is the ECM disorder caused by NP cell damage [49]. Collagens (mainly type I and II collagens), proteoglycans, and elastins comprise the main components of the ECM. Among them, aggrecan is the main proteoglycan found in IVDs and is a material basis of ECM homeostasis [50]. Matrix metalloproteinases (MMPs) [51], mainly including MMP1, 2, 3, 7, 8, 10, and 13, as well as a disintegrin and metalloproteinase with thrombospondin motifs (ADAMTSs) [52], mainly including ADAMTS-1, 4, and 5, largely contribute to the degradation of the ingredients outlined above. The critical step of ECM disorder is proteoglycan degradation, accounting for the decreased hydration capacity of IVDs. This feature reduces the compression capacity and easily causes the expansion of the NP, resulting in cascading changes [53]. Divided into the outer, middle, and inner layers, the AF is a lamellar structure rich in collagens which can disperse mechanical stress to maintain the flexibility and stability of the spine. The nerve fibers and blood vessels distributed in the AF can transport nutrients to maintain the metabolism balance in the NP [54]. Apart from providing nutrients, the AF can disperse stress to prevent the radial disc bulge of the NP. At the same time, the additional stress caused by NP denaturation adds a burden to the outer layer of the AF. Specifically, the interval between the AF and the NP is vague, and the gaps between each layer are increased or even broken, eventually accelerating the loss of proteoglycan and type I collagen fibers [55]. The destruction of this structure leads to a functional change. In the AF, this is manifested by a decrease in antistretching ability and increased angiogenesis, triggering neuropathic pain and further impairing tissue homeostasis [56]. Likewise, the EP has the function of fixing and dispersing mechanical stress. In addition, the EP also performs as a translucent barrier between the avascular IVD and highly vascularized vertebrae, transporting oxygen, glucose, and metabolites through diffusive movement [57]. During the IDD process, EP becomes thinner and more calcified [58], resulting in a vicious circle as a result of hindered solute diffusion. Eventually, inflammation, hypoxia, acidity, and nutrient deprivation can be witnessed in the microenvironment of IDD. Biochemical and pathological changes, as noted above, aggravate the loss of proteoglycans, inhibit collagen cross-linking, and form osteophytes [59]. T1-weighted images show the diagnostic indicator of hyperintensity on the torn AF, whereas T2-weighted images display low signals of atrophy and dehydration in the NP region [60].

## 4. Roles of Pyroptosis in IDD

### 4.1. Pyroptosis Triggers Cell Death in IDD

Cell death, especially NP cell death, is one of the most critical factors triggering IDD. It leads to a loss of cell function and a decline in nutrient synthesis and repair capability and, moreover, increases inflammation and oxidative stress [7]. All of this aggravates the degenerative cascade in IVDs. A positive correlation between NLRP3 expression and the degenerative score was found in 45 clinical samples [61]. This result demonstrated that pyroptosis is widely involved in IDD progression, which is mainly induced by NLRP3 inflammasome (Figure 2).

Besides directly causing cell death, pyroptosis and other programmed-cell-death pathways are jointly cross-regulated in the complex and harsh degenerative environment. It was found that, after lumbar spine instability surgery, the height of the IVD was reduced in mice, and fissures and folds were formed between the AF layers, with osteophyte formation in the EP. The TUNEL assay and pyroptosis-related protein detection showed that both apoptosis and pyroptosis in the IDD-model group were potentiated [62]. Studies on anti-inflammatory protein A20 further revealed that when pyroptosis was inhibited, the expression of inflammatory cytokines coupled with apoptosis was also markedly reduced [63]. The above results suggest that there is an interplay between apoptosis and pyroptosis in IDD. The molecular mechanism behind the interplay may be explained by PANoptosis [64], which is characterized by pyroptosis, apoptosis, and necroptosis. Li et al. [65] first discovered PANoptosis, consisting of pyroptosis and apoptosis, in IDD lesions. They used different concentrations of TNF-α to treat NP cells and found that both pyroptosis and apoptosis were upregulated. However, pyroptosis was predominant at low concentrations, whereas apoptosis was predominant at high concentrations, implying that the major type of cell death may vary during different periods of IDD progression [65]. This interconverting relationship may be partially explained by the altered concentrations of inflammatory mediators in the IDD microenvironment and the activation of the CASP3-GSDME pathway in the crosstalk of pyroptosis and apoptosis [66]. Moreover, the relationship between pyroptosis and autophagy is also intriguing. In bromodomain-containing protein 4-inhibited rat NP cells, enhanced autophagy was able to reduce pyroptosis to relieve degenerative development. Whether autophagy enhancement is linked with attenuated pyroptosis requires further investigation; however, it has been demonstrated that the NF-κB signaling pathway is involved [67]. Similarly, one study found a tendency for autophagic vacuoles to increase with higher microtubule-associated protein 1 light chain 3 (LC3)-II/LC3-I ratios in NLRP3 knockout NP cells, demonstrating that pyroptosis and autophagy are antagonistic to each other [68]. Moreover, some researchers have tried to explore the mechanism underlying this phenomenon. One study pointed out that autophagy protects against LPS-induced NP cell pyroptosis. The explicit regulatory relationship between them depends on SQSTM1/p62, one of the autophagy-related proteins, co-localizing with GSDMD-N. It degraded GSDMD-N through the autophagy–lysosome pathway, thereby attenuating pyroptosis [69]. In conclusion, the interplay between pyroptosis and other programmed cell deaths of NP cells in IDD currently remains questionable. It is suggested that subsequent experiments focus on this aspect, which may have a significant role in revealing the progression of IDD.

### 4.2. Pyroptosis Provokes ECM Disorder in IDD

The composition, synthesis, and reconstruction of the ECM are extremely important for the stability of the IVD internal environment. When the ECM metabolism is severely disturbed, the hydration capacity of the IVD decreases, becoming stiffer and more fragile, thus affecting biological function [70]. Previous observations have shown that pyroptosis can induce ECM degradation. For example, the downregulation of nicotinamide phosphoribosyl transferase blocks the priming phase of NLRP3 through the NF-κB and MAPK pathways, reducing the degradation of aggrecan and collagen II [71]. Cholesterol accumulates in IDD, showing a repression of aggrecan and collagen II, as well as an elevation of MMP13 and ADTAMTS5. This is related to the activation of endoplasmic reticulum (ER) stress, an activating phase of NLRP3 [72]. Although their signals differ, both of them are DAMPs, which not only activate pyroptosis in NP cells, but also reduce ECM synthesis and increase ECM degradation. In addition to directly disrupting the balance of ECM synthesis and decomposition by causing NP cell death, the inflammatory mediators and cellular contents released by pyroptosis are largely involved in ECM disorder as well. NP cells are instrumental in the ECM metabolism [73]. When healthy NP cells co-culture with pyroptotic ones, the expressions of MMP13, ADAMTS4, and ADAMTS5 are upregulated [22]. The fragment products they hydrolyze damage the stress-resistance capability of the ECM and provoke secondary inflammatory responses [74]. TNF-α and IL-1β are acknowledged factors driving the synthesis and release of MMPs and ADAMTSs [75], which are also in close contact with pyroptosis. The mediators mentioned above may result in ECM degradation caused by pyroptosis. Additionally, this phenomenon is also found in Modic EP changes and degenerated AF tissues and is thus not limited to NP cells [61,62].

In addition, ECM disorder also aggravates pyroptosis. Previous research found that anaerobic glycolysis in degenerated NP cells creates an acidic environment which activates ROS/NLRP3/caspase-1-mediated pyroptosis [23]. However, one study reached the opposite conclusion. With increased acidification, ECM degradation may be aggravated by exogenous factors rather than the decreased NLRP3 in NP cells [76]. On the other hand, the accumulation of metabolites, such as advanced glycation end products, hardens collagen fibers and facilitates the assembly of NLRP3 [77]. This shows promise in regard to restoring ECM disturbances and delaying IDD progression by inhibiting pyroptosis. An injection of VX-76, a caspase-1 inhibitor, can prevent the replacement of proteoglycans by fibrous tissue and slow the aging and fibrosis of the IVD effectively [69].

### 4.3. Pyroptosis Induces Secondary Inflammation in IDD

Increased secondary inflammatory cytokines are regarded as a significant feature of symptomatic IDD [78]. The inner region of the IVD lacks vascularized structures and is isolated from the host immune system. NP cells usually lack immune tolerance but are capable of phagocyting and secreting inflammatory factors. When exposed to an inflammatory environment, they will produce a solid autoimmune and inflammatory cascade [79,80]. The inflammation in IDD manifests in three ways, which are partly assigned to pyroptosis.

Changes in IVD structure and physiology lead to the production of inflammatory factors, which is the most distinctive feature of the first stage [81]. Characterized by a typical inflammatory cell death mode, pyroptosis unleashes proinflammatory factors in IVD cells either directly or indirectly [17]. As a direct product of pyroptosis, IL-1β is recognized as a key cytokine involved in discogenic back pain [82]. At the same time, when exposed to IL-1β, the expression of inflammatory factors such as IL-6 and IL-8 is significantly elevated [83]. It is not uncommon to see that some substances in IDD can trigger an inflammatory reaction by inducing pyroptosis. Extracellular ATP (eATP) is a representative co-activator of NLRP3 and P2X7R from the ATP-gated ion channel family. When strongly stimulated, P2X7R transfers to the cytoplasm to co-locate with NLRP3 and release transforming growth factor-β (TGF-β) and IL-1. It is also related to IVD inflammation [84]. However, whether eATP causes inflammation directly or through pyroptosis remains to be further elaborated. Although IDD is mainly characterized by aseptic inflammation at this stage, infectious factors have gradually attracted attention in recent years. Among them, propionibacterium acnes account for 13% to 44% of the causes of IDD [85]. It was reported that propionibacterium acnes could induce the pyroptosis of NP cells through the ROS-NLRP3 signal pathway. Pyrolytic products aggravate the aggregation of inflammatory factors in IDD [22]. On the contrary, an injection of the NLRP3 inhibitor MCC950 successfully improves the inflammatory environment of IDD [86], suggesting that targeted pyroptosis may be a therapeutic strategy to prevent infection factors from aggravating the inflammatory progress of IDD. However, to date, in-depth insights into the specific antigens of the infectious factors inducing pyroptosis are still lacking. In addition, although pyroptosis is involved in a heightened inflammatory response in IDD, whether pyroptosis is the initial cause of IDD inflammation deserves further discussion.

The infiltration of leukocytes and the ingrowth of blood vessels and nerves are second-stage events. The results of this incremental process are closely related to the third stage, so we will expound them together. The third stage is identified by the pain symptoms caused by the sensitization of nerve endings and nociceptive neuron infiltration [81]. Sensory nerve fibers and blood vessels are usually distributed in the outer layer of AF tissue for nutrient transport [87]. In the mouse lumbar-spine-instability model, vascular bundles and sensory nerves grew in the inner and outer regions of the AF, with calcitonin gene-related peptide (CGRP)-positive cells increasing [62]. This may be attributed to the fact that cytokines released by pyroptosis aggravate the loss of proteoglycans in the ECM and promotes angiogenesis [88]. CGRP is an important nociceptive neurotransmitter for controlling inflammation and pain [89]. Additionally, a large number of inflammatory factors released by cell rupture constantly stimulate sensory nerves, which is also a significant pain factor. A study utilizing NF-κB inhibitor Bay11-7082 on the lumbar-disc-herniation rat model found that the assembly of NLRP3 was blocked, and CGRP and pain-related behaviors such as allodynia and heat pain thresholds were ameliorated. This suggests that NF-κB, the standard signal of pain and pyroptosis, is involved in relieving neuropathic pain [90]. Moreover, although the relationship between ion channels and inflammation has not been fully explained, Ca^2+^-dependent Pizol1 channels activated by mechanical stretching [91] and ASIC3 channels marked by acid sensing [23] can promote pyroptosis, which may influence the sensitization of pain in the IVD later [92].

## 5. Therapeutic Strategies Targeting Pyroptosis and NLRP3 Inflammasome in IDD

### 5.1. Exogenous Drugs

Exogenous drugs mainly refer to plant-derived natural compounds and chemical compounds that affect the human body through external administration. Thanks to their low toxicity, rich sources, and extensive pharmacological effects, plant-derived natural compounds have gradually become a hot topic in drug discovery [93]. Under the guidance of precise chemical structures, chemical compounds are essential for developing new chemical skeletons or tapping the activity of old drugs [94]. In IDD, exogenous drugs mainly target the NLRP3 inflammasome to inhibit pyroptosis (Table 1). As mentioned above, activating the NLRP3 inflammasome requires priming and activating phase signals. The activation of the transcriptional regulator dominates the priming signal to increase the expression of the NLRP3 inflammasome [33]. TLR4 is the most representative molecule of the Toll family, and can recognize ligand-independent signaling to recruit adapter proteins MyD88, TIRAP, and TRAM, and ultimately participate in the robust activation of NF-κB to increase NLRP3 transcription [95]. In the H_2_O_2_-mediated-degeneration model, ganoderic acid significantly inhibited the expression of TLR4 and NLRP3. However, it still remains uncertain whether ganoderic acid directly acts on TLR4 or triggers the secondary reduction in TLR4 by scavenging H_2_O_2_ [96]. In addition to H_2_O_2_, high-mobility group box-1 protein (HMGB1) and DAMPs also act on TLR4 receptors [97]. Magnoflorine reduced HMGB1 and blocked the recruitment of the Myd88 adapter, and then inhibited the NF-κB/NLRP3/caspase-1 pathway [98]. The released HMGB1 induced by LPS-medicated noncanonical inflammasome-induced pyroptosis [99] propagates the inflammatory response further through phagocytosis, and performs as a DAMP, finally activating caspase-1-dependent pyroptosis [100]. Detecting caspase-11 in this model may further reveal the role of magnoflorine. Oseltamivir [101] and Bay11-7082 [90] can also inhabit the NLRP3 inflammasome through the NF-κB pathway to delay the progression of IDD.

The second signals are related to various intracellular signal disorders concerning ionic flux and mitochondrial dysfunction [33]. Atorvastatin, primarily used to treat cardiovascular and inflammatory diseases, is found to reduce cholesterol content, inhibit inflammatory infiltration, and relieve ER stress to remodel the ECM in IDD [72]. Lipid disturbance caused by cholesterol increases ER pressure and is associated with mitochondria releasing Ca^2+^ [102]. An acid environment can also promote Ca^2+^. Under the intervention of paeoniflorin, the dropped PH and Ca^2+^ release was well-controlled, alleviating NLRP3-mediated pyroptosis and significantly improving the MRI pathological score of IDD [103]. ROS is also a key molecule for NLRP3 activation. Gamma oryzanol is a dietary supplement that can reduce the intracellular ROS and mitochondrial superoxide levels in NP cells. Its mechanism is possibly related to a reduction in thioredoxin-interacting protein (TXNIP), an ROS-sensitive protein that promotes NLRP3 oligomerization [104,105]. Morin [106] and honokiol [107] are potential small molecules for treating IDD through the TXNIP/NLRP3 pathways.

Although great strides have been made to explore exogenous components to delay IDD by regulating pyroptosis, explorations of their metabolic distribution and therapeutic doses are still lacking. Furthermore, almost no in vivo experiments have been carried out on humans thus far. Long-term toxicity and the mode of administration may be challenging issues for subsequent development.

**Table 1 biomolecules-12-01804-t001:** Mechanisms of exogenous compounds for IDD treatment by inactivating pyroptosis.

Type	Compound	Dose	Model Type	Mechanism					References
				Pathway	Cell Death	ECM	Inflammation	Others	
Plant-derived natural compounds	Ganoderic Acid A	08 μM	Rat NP tissue (in vivo/in vitro)	Inactivated TLR4/NLRP3	Suppressed apoptosis	Suppressed MMP3, MMP13, ADAMTS4, ADAMTS5. Upregulated Col II, aggrecan	Alleviated IL-6, IL-1β, TNF-α	Reduced oxidative stress by restoring GSH, SOD, GPX	[96]
Plant-derived natural compounds	Magnoflorine	100 μg/mL	Human NP cells (in vitro)	Inactivated HMGB1/ MyD88/NF-κB/NLRP3	Suppressed apoptosis	Suppressed MMP3, MMP13, ADAMTS4, ADAMTS5. Upregulated Col II, aggrecan	Alleviated HMGB1, IL-1β, IL-6, TNF-α, IL-18	Alleviated “M1” polarized macrophage	[98]
Plant-derived natural compounds	Paeoniflorin	20 mg/kg	Rat NP tissue (in vivo/in vitro)	Inactivated IL-1β/NLRP3	Suppressed pyroptosis	Suppressed MMP3, MMP13. Upregulated Col II, aggrecan	Alleviated IL-1β, IL-18	Reduced calcium concentration	[103]
Plant-derived natural compounds	Gamma oryzanol	40 μM	Rat NP tissue (in vivo/in vitro)	Inactivated IL-1β/NF-κB/NLRP3	Suppressed pyroptosis	Suppressed MMP13, ADAMTS5. Upregulated Col II, aggrecan	Alleviated IL-1β, IL-18	——	[104]
Plant-derived natural compounds	Morin	50 mg/kg	Rat NP tissue (in vivo/in vitro)	Inactivated TXNIP/ NLRP3	Suppressed pyroptosis	——	Alleviated IL-1β, IL-18, TNF-α	——	[106]
Plant-derived natural compounds	Honokiol	40 μM	Rat NP tissue (in vivo/in vitro)	Inactivated NF-κB, JNK, TXNIP/NLRP3	Suppressed apoptosis	Suppressed MMP3, MMP13, ADAMTS4, ADAMTS5. Upregulated Col II, SOX9	Alleviated IL-6, COX-2, iNOS	Reduced oxidative stress by restoring MDA, SOD, GPX	[107]
Chemical compound	Bay11-7082	5 mg/kg	Rat NP tissue (in vivo/in vitro)	Inactivated NF-κB/NLRP3	——	——	Alleviated IL-1β, IL-18. Downregulated CGRP in DRG neurons	——	[90]
Chemical compound	Atorvastatin	20 μM	Rat NP tissue (in vivo/in vitro)	Inactivated NF-κB/NLRP3	Promoted autophagy. Suppressed pyroptosis	Suppressed MMP3, MMP13, ADAMTS4, ADAMTS5. Upregulated Col II, aggrecan	Alleviated IL-1β, IL-18, TNF-α	Alleviated SREBP1 mediated cholesterol-induced pyroptosis and ER stress	[72]

### 5.2. Endogenous Small Molecules

Endogenous small molecules are substances with physiological functions and biological activities that exist naturally in human or mammalian bodies. They have strong pharmacological effects and few side effects [108]. Endogenous small molecules from a wide range of sources affect IDD through multiple pyroptosis-related signaling pathways. Nicotinamide phosphoribosyl transferase inhibitor [71] and platelet-derived growth factor BB [109] can inhibit NLRP3-mediated pyroptosis and play a protective role in ECM by NAMPT/MAPK/NF-κB and MAPK/PI3K/ERK, respectively. Maresin 1 [110], melatonin [111], cortistatin [112], platelet-rich plasma [113], MFG-E8 [114], A20 [63], lipoxin A4 [115], and SIRT1 [116] can reduce antioxidative stress and inhibit inflammatory factors via the NF-κB, NAMPT/MAPK, Keap1-Nrf2, Nrf2/TXNIP, and JNK1/Declin-1/PI3KC3 pathways. The specific mechanisms are shown in Table 2. Although increasing attention has been paid to the development of endogenous small molecules, the direct administration of endogenous molecules or the induction of the expression of endogenous molecules through activators is still an issue worthy of consideration. Additionally, if administered directly, how to prevent degradation as a result of pharmacokinetics is also an inevitable question.

### 5.3. Stem Cells and Bioengineering

Due to the special anatomical structure and complex physiological environment of the IVD, exogenous cells struggle to survive, and the efficiency of the drug delivery route is poor [117]. At present, stem cells and bioengineering are used to overcome the issues related to physiological structure and mechanical characteristics, opening new therapeutic avenues for IDD treatment [118]. Stem cells can release exosomes, which are extracellular vesicles 50–100 nm in diameter. They have a low risk of rejection and efficient drug administration [117]. It has been proven that IDD can be alleviated by inhibiting pyroptosis-related signaling pathways through exosomes. Exosomes derived from human umbilical cord mesenchymal stem cells deliver miR-26a-5p to inhibit m6A methylation in NLRP3-medicated pyroptosis by reducing methyltransferase METTL14 in NP cells [119]. Mesenchymal stem-cell-derived exosomal miR-410 [120] and platelet-plasma-derived exosomal miR-141-3p [113] can also target NLRP3 as efficient therapy options to treat IDD. On the basis of the above results, how to improve the targeting efficiency and durability of exosomes is worth discussing. Xing et al. [121] designed an injectable exosome-functionalized extractive-matrix hydrogel by fixing the exosomes extracted from adipose-tissue-derived mesenchymal stem cells on the ECM gel. This device not only directly replenished the ECM, but also continuously increased the loading rate of exosomes to maintain the stability of the ECM by inhibiting pyroptosis. Apart from exosomes, extracellular vesicles (EVs) generated by stem cells are also a good therapeutic choice. Zhang et al. [122] constructed a Cavin-2-modified EV system derived from mesenchymal stem cells, which was loaded with NLRP3 inflammasome antagonist peroxiredoxin-2. The system increased the uptake and permeability rate of EV under TNF-α-agglomerated proinflammatory conditions in IDD. Esterase-responsive ibuprofen nanomicelle premodified embryo-derived nucleus pulposus progenitor cells show outstanding advantages. Esterase-responsive ibuprofen nanomicelles can release ibuprofen into the acidic environment of IDD, which improves the microenvironment of IDD through inhibiting COX2/NF-κB/caspase-1-mediated pyroptosis [123]. Furthermore, ensuring AF cell resistance against NP stress is another feasible research direction. Zhang et al. [124] prepared a fabrication of a polylactide-glycolide/poly-ε-caprolactone/dextran/plastrum testudinis extract (PTE) composite with an anti-inflammation nanofiber membrane. Loaded with PTE, it inhibited NLRP3 and its upstream and downstream signaling pathways and promoted AF cell regeneration.

The therapeutic approaches designed by bioengineering methods targeting pyroptosis and the NLRP3 inflammasome in IDD have broad prospects. In the future, exploring different methods of transportation and bioactive molecules loaded with biomaterials tailored to pyroptosis could be a potential direction. At the same time, it could be promising for the follow-up bioengineering research on IDD to focus on other markers of pyroptosis, such as CASP1, GSDMD, and ASC.

## 6. Summary and Outlook

Pyroptosis, characterized by the secretion of inflammatory factors and cell destruction, is a new type of cell death that has attracted much attention in the last decade. It has been proven to be associated with IDD. Specifically, pyroptosis is involved in the microscopic progression of IDD by triggering cell death, ECM disorder, and secondary inflammatory responses. These three processes can also further influence pyroptosis, which eventually develops into clinically typical structural destruction and the progressive dysfunction of the IVD, severely reducing patient quality of life.

Although the relationship between pyroptosis and IDD has been recognized, there are still deficiencies. First, a large number of homogenization studies have hindered the search for new mechanisms. The research on IDD primarily focuses on NLRP3 inflammasome-mediated pyroptosis, and little is known about NLRP1/4/9 and AIM2. NLRP1, AIM2 and GSDME are usually reduced or missing in livestock such as pigs, cats, and cattle, making them more resistant to bacterial invasion [125]. The above pyroptotic components may partly explain the reason why humans are more vulnerable to IDD compared to other vertebrates. In other words, the animal-model in vivo experiments should be chosen with caution; otherwise, the results may not reflect the real situations in humans. Secondly, considering the three anatomical structures that coordinate the function of IVDs, the current research on pyroptosis in IDD is chiefly limited to the NP, and the doubts concerning the AF and the EP are yet to be investigated. Moreover, a considerable number of studies have concentrated on molecular biology, but only a few researchers have evaluated changes in the macrostructure of the IVD, which also hinders the foundation of clinical biomarkers. Additionally, no IDD model can replicate the human IDD phenomenon; most experiments at present lack clinical validation and are not sufficiently convincing. The absence of application to the human body also limits our understanding of the required dosage and side effects of pyroptosis therapy. Apart from solving this dilemma, the next step could be to focus on the interaction of pyroptosis with other forms of death in IDD. In conclusion, the inhibition of pyroptosis is a promising strategy for treating IDD.

## Figures and Tables

**Figure 1 biomolecules-12-01804-f001:**
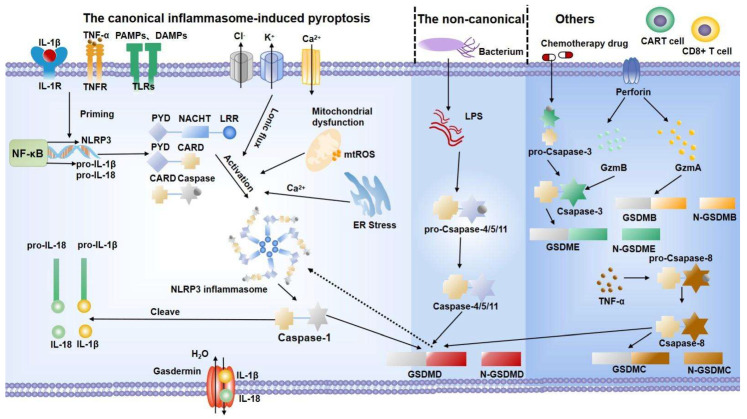
Pyroptosis pathway. The canonical inflammasome-induced pyroptosis pathway (taken NLRP3 inflammasome pathway) begins with the interaction between receptors and ligands. Then, the NF-κB signal is activated and increases the expression of the NLRP3 inflammasome. Meanwhile, activation signals, including various disturbances, improve the assembly of the NLRP3 inflammasome. Then, the assembled NLPR3 activates caspase-1 to mature IL-18 and IL-1β. The noncanonical inflammasome-induced pyroptosis is mediated by LPS to activate caspase-4/5/11 to cleave GSDMD. In other ways, different cells activate the gasdermin family through caspase-dependent or caspase-independent pathways, which eventually lead to cell expansion, membrane perforation, and the release of cytoplasmic content. IL: interleukin; TNF-α: tumor necrosis factor-α; PAMPs: pathogen-associated molecular patterns; DAMPs: damage-associated molecular patterns; TLRs: toll-like receptors; ROS: reactive oxygen species; ER: endoplasmic reticulum; LPS: lipopolysaccharide; GSDMD: gasdermin D; GzmA: granzyme A; GzmB: granzyme B; CAR-T: chimeric antigen receptor T.

**Figure 2 biomolecules-12-01804-f002:**
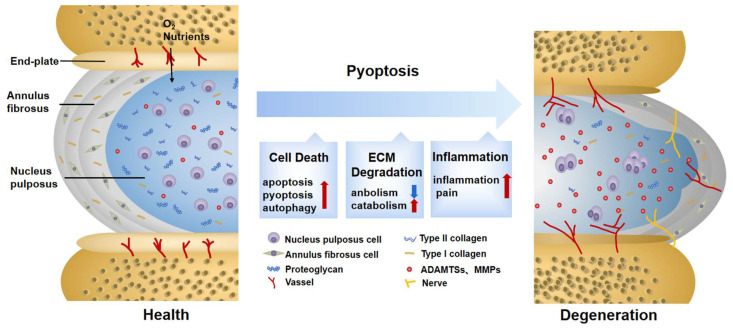
Changes in normal and degenerative intervertebral discs. The intervertebral disc is composed of nucleus pulposus (NP), annulus fibrosus (AF), and endplate (EP), together constituting a closed buffer system against stress. While in IDD, cell death, ECM degradation, and secondary verification aggravate the vicious cycle, which is closely associated with pyroptosis. ECM: extracellular matrix; ADAMTS: a disintegrin and metalloproteinase with thrombospondin motifs; MMPs: matrix metalloproteinases.

**Table 2 biomolecules-12-01804-t002:** Mechanisms of endogenous molecules for IDD treatment by inactivating pyroptosis.

Compound	Dose	Model Type	Mechanism					References
			Pathway	Cell Death	ECM	Inflammation	Others	
Melatonin	1000 μM	Human (in vitro)/Rat NP tissue (in vivo/in vitro)	Inactivated NAMPT/MAPK/NF-κB/NLRP3	Suppressed pyroptosis	Upregulated Col II, aggrecan	Suppressed IL-1β, IL-18, TNF-α	——	[71,111]
Platelet-Derived Growth Factor-BB	50 ng/mL	Rat NP tissue (in vivo/in vitro)	Inactivated MAPK/PI3K/AKT/NLRP3	Suppressed pyroptosis and apoptosis	Suppressed MMP3, MMP9, ADAMTS4, ADAMTS5. Upregulated Col II, aggrecan	Alleviated IL-1β, IL-18	——	[109]
Cortistatin	50 μg/mL	Human (in vitro)/Mice NP tissue (in vitro)	Inactivated NF-κB/NLRP3	Suppressed apoptosis	Suppressed MMP13, ADAMTS5. Upregulated Col II, aggrecan	Alleviated IL-1β, TNF-α	Alleviated respiratory chain. Suppressed mitochondrial ROS generation	[112]
Maresin 1	100 ng	Rat NP tissue (in vivo/in vitro)	Inactivated NF-κB/NLRP3	Suppressed pyroptosis	——	Alleviated IL-1β, IL-18, TNF-α	Alleviated mechanical allodynia. Improved radicular pain	[110]
Platelet-rich plasma	——	Mice NP tissue (in vivo/in vitro)	Activated the Keap1-Nrf2	Suppressed pyroptosis	——	Alleviated IL-1β, IL-18, TGF-β, IL-6	Alleviated ROS production	[113]
A20	——	Rat NP tissue (in vivo/in vitro)	Inactivated NF-κB/NLRP3	Promoted mitophagy. Suppressed pyroptosis and apoptosis	——	Alleviated iNOS, COX2, TNF-α, IL-1β, IL-6, IL-18	Stabilized mitochondrial dynamics. Inhibited collapse of mitochondrial membrane potential and ROS	[63]
MFG-E8	100 ng/ml	Rat NP tissue (in vivo/in vitro)	Inactivated Nrf2/TXNIP/NLRP3	Suppressed pyroptosis	Suppressed MMP3, ADAMTS5. Upregulated Col II, aggrecan	Alleviated IL-1β, IL-18	Suppressed ROS and mitochondrial dysfunction	[114]
Lipoxin A4	10 µL, 100 ng	Rat NP tissue (in vivo/in vitro)	Activated JNK1/Beclin-1/PI3KC3/NLRP3	Promoted autophagy	——	Decreased TNF-α, IL-1β, IL-18. Increased IL-4, IL-10, TGF-β	Ameliorated the pain threshold	[115]

## Data Availability

Not applicable.

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
