# Peer review of "Role of Pyroptosis in Intervertebral Disc Degeneration and Its Therapeutic Implications"

_biomolecules, 2022, doi:10.3390/biom12121804_

Round 1
Reviewer 1 Report
This is a good review covering many aspects of pyroptosis that has been discovered in the last decade. Some suggestions are: a) include discussion that pyroptosis is currently understood to often occur along with other cell death pathways (the concept of PANoptosis is there). The inflammation assigned to pyroptosis may be due to multiple death platforms getting activated with pyroptotic signaling. b) discern causes of inflammation that impact the spine, where pyroptosis may contribute. One example is discitis. Discitis is spinal inflammation due to infection that can lead to IDD. It will be great to see some discussion about infectious causes of IDD as pyroptosis is primarily activated by infection. c) IDD is observed in many vertebrates including cats and dogs who lack several components of the pyroptotic and inflammatory signaling. Vet treatment will need to consider those aspects. It is important to point out that contributions from pyroptosis to IDD may be species specific and primarily a human phenomenon. d) formatting (inconsistent font) issues need to be fixed.
Author Response
Dear Reviewer 1,
Thank you for signing your review report. And thank you for your comments and suggestions which help us to improve the academic rigor of our manuscript. The point-by-point response letter is uploaded, please see the attachment. All modifications of the revised manuscript that are related to the content are marked up using "Track Changes". We sincerely hope that the revised manuscript will meet your satisfaction.
Kind regards
Mr. Yuxuan Yang
Nov. 27th

Reviewer 2 Report
This review summarises the current knowledge with regards to pyroptosis and how this type of cell death affects the progression of IDD. The text is well written and provides the essential information about the molecular mechanisms that govern pyroptosis and the different activation pathways. They also include a brief description of the IDD pathogenesis along with some therapeutic options inhibiting pyroptosis which results in alleviating the symptoms of IDD. This is a good review article; however, the authors need to address some points in order to improve the quality of the manuscript.
· There are no in-text references for the figures. Why the figures are provided as supplemental material and not in the main text?
· Line 105, some examples of chemotherapeutic drugs must be included.
· Line 127, which are these proteoglycans in IVD that are being degraded? In section 3, there is no reference to MMPs, ADAMTSs and other proteolytic enzymes which are responsible for ECM degeneration in IDD.
· Section 4.1 needs a better flow. It involves pyroptosis, PANoptosis and autophagy but is difficult to follow the concept.
· Lines 182-184, how Cholesterol and nicotinamide phosphoribosyltransferase reduce ECM synthesis and increase ECM degradation?
· Lines 220-227, there is no connection of this part to the other content of the section. It should be removed.
· Are there any in vivo studies in animal models that inhibit pyroptosis to tackle IDD? This should be described and info provided in a separate table.
· Section 5.3, the relation to pyroptosis is not obvious. This section should be rewritten. The authors also implicate miRs but do not provide further details. It would be better to remove these references as the manuscript tends not to be focused. Line 22, what does “terrible conditions” mean? Lines 26-27, how ibuprofen inhibits pyroptosis?
· Finally, the text needs extensive and very careful editing as some errors occur.
Author Response
Dear Reviewer 2,
Thank you for signing your review report. And thank you again for your comments and suggestions which help us to improve the academic rigor of our manuscript. The point-by-point response letter is uploaded, please see the attachment. All modifications of the revised manuscript that are related to the content are marked up using "Track Changes". We sincerely hope that the revised manuscript will meet your satisfaction.
Kind regards
Mr. Yuxuan Yang
Nov. 27th

Reviewer 3 Report
· The overall write-up of the manuscript is completely non-scientific.
· Sentences are improperly written.
· Most of the information is reflecting on how NLPR3 promotes pyroptosis which is irrelevant to the topic.
Abstract:
· line 13 states that” pyroptosis is recently discovered” however, this statement is inappropriate.
· line 15 vaguely describes that “pyroptosis is characterized by rupture of plasma membrane”. However, no cell type is mentioned.
Introduction:
· line 33 states secondary inflammation without elaboration.
· line 48 mentions ’a broad set of studies” with only one citation. Therefore more references need to be cited.
· This article contains words like “ In this way, whats more, another argument, etc’’ which reflect spoken English rather than scientific terminologies.
· There is a lack of references supporting multiple statements
· Subheadings 4.2 and 4.3, lines no 181-184 are completely irrelevant to their headings
Subheading 5.2 does not address any pharmacological agent but mentions only plant and animal-derived products falling under the category of exogenous compounds
Subheading 5.3: The description is reflecting the role of stem cells in reducing disc degeneration however it does not elaborate its effects on pyroptosis.
Author Response
Dear Reviewer 3,
Thank you for signing your review report. And thank you again for your comments and suggestions which help us to improve the academic rigor of our manuscript. The point-by-point response letter is uploaded, please see the attachment. All modifications of the revised manuscript that are related to the content are marked up using "Track Changes". We sincerely hope that the revised manuscript will meet your satisfaction.
Kind regards

Round 2
Reviewer 2 Report
The authors sufficiently addressed the comments. This is a good review and can be accepted for publication in this revised form.